# Clinical Sensor-Based Fall Risk Assessment at an Orthopedic Clinic: A Case Study of the Staff’s Views on Utility and Effectiveness

**DOI:** 10.3390/s23041904

**Published:** 2023-02-08

**Authors:** Maria Ehn, Annica Kristoffersson

**Affiliations:** School of Innovation, Design and Engineering, Mälardalen University, Box 883, 721 23 Västerås, Sweden

**Keywords:** falls, healthcare, hospital, prevention, fall risk, assessment, inertial sensors, wearable sensors, technology adoption

## Abstract

In-hospital falls are a serious threat to patient security and fall risk assessment (FRA) is important to identify high-risk patients. Although sensor-based FRA (SFRA) can provide objective FRA, its clinical use is very limited and research to identify meaningful SFRA methods is required. This study aimed to investigate whether examples of SFRA methods might be relevant for FRA at an orthopedic clinic. Situations where SFRA might assist FRA were identified in a focus group interview with clinical staff. Thereafter, SFRA methods were identified in a literature review of SFRA methods developed for older adults. These were screened for potential relevance in the previously identified situations. Ten SFRA methods were considered potentially relevant in the identified FRA situations. The ten SFRA methods were presented to staff at the orthopedic clinic, and they provided their views on the SFRA methods by filling out a questionnaire. Clinical staff saw that several SFRA tasks could be clinically relevant and feasible, but also identified time constraints as a major barrier for clinical use of SFRA. The study indicates that SFRA methods developed for community-dwelling older adults may be relevant also for hospital inpatients and that effectiveness and efficiency are important for clinical use of SFRA.

## 1. Introduction

Fall accidents are a major threat to the health of older adults, resulting in injuries and even premature death. The frequency of falls increases with age and frailty level [1]. Approximately 30% of all community-dwelling people ≥65 years [2,3,4] and 30–50% of older adults living in long-term care institutions [5] fall every year. Moreover, falls are the most common adverse event reported in hospitals [6] with reported fall rates ranging from 1.3 to 21 falls/1000 patient days [7,8].

Approximately 10% of all falls occurring among older adults cause severe injury, most commonly fractures [9]. The number is higher for patients admitted to hospitals (approximately 35% [10,11]) or acute hospitals (51%) [12]). Previous studies have identified that approximately 1–2% of all in-hospital falls result in fractures [11,13,14]. In-hospital falls are associated with longer hospital stays, higher hospital-related costs, and higher rates of discharge to institutional care [15,16,17].

Although fall prevention has been relatively well-studied among older adults living in the community, research in care facilities and hospitals is more limited. A Cochrane Review on fall prevention interventions for older adults [18] included 159 trials (79,193 participants) while the same type of review on interventions for older adults [19] included 71 trials (40,374 participants) in care facilities and 24 trials (97,790 participants) in hospitals. Cameron et al. [19] identified that multifactorial interventions may reduce the fall rate in hospitals, and LeLaurin and Shorr [6] emphasized that there is an urgent need for well-designed research studies on hospital fall prevention. Oliver et al. [20] identified that: (1) the included components differ widely among multifactorial fall prevention interventions; (2) fall risk assessment (FRA) tools, called fall prediction in [20], are only used in half of the successful trials; and (3) spending extra resources on FRA rather than on intervention efforts in high-cost trials is not associated with a guaranteed success. Oliver et al. [20] discuss potential risks and benefits of providing more general fall prevention efforts to all patients versus tailoring efforts towards identified high-risk patients. Examples of FRA methods include Morse Falls Scale [21], Tinetti Performance Oriented Mobility Assessment (POMA) [22], Physiological Profile Approach [23], and the STRATIFY Score [24]. The Morse Falls Scale and STRATIFY Score, which have been subjected to external validation in different hospital settings and inpatient groups [25,26,27], showed relatively low specificity and sensitivity, and an even lower positive predictive value [20]. A systematic review and meta-analysis [28] identified that the Morse Falls Scale and STRATIFY Score may be useful in particular settings, but that widespread adoption of either of them is unlikely to generate significantly higher predictive accuracy than that of nursing staff clinical judgment. The nurses’ key strategies to ensure patient safety related to falls include assessment, monitoring, and communication [29].

There is a major need for clinical tools that are easy-to-use and generate objective, accurate, and quantitative risk assessment in clinical settings [30,31]. This has generated an interest in sensor-based FRA (SFRA) which base the assessments on signals from wearable sensors monitoring an individual’s motions while performing specific assessment tasks [32]. So far, SFRA has mainly been evaluated in research studies [32,33] and several systematic reviews of SFRA literature [31,34,35,36,37] have identified a large variation in these tools, measured parameters, assessment tasks, sensor positions, fall risk models, etc. Hence, the need for research that clinically identifies meaningful SFRA methods and addresses the gap between functional evaluations and user experience has been highlighted [37]. User experience can be defined as “A person’s perceptions and responses that result from the use or anticipated use of a product, system or service”, p.4 [38]. The concept of user experience focuses on design and prediction of adoption of new technologies (technology acceptance) [39]. Several models describe factors that influence users’ acceptance of new technologies, e.g., the Technology Acceptance Model (TAM), which is based on the Theory of Reasoned Action [40]. TAM postulates that the two user variables “perceived usefulness” and “perceived ease of use” are determinants of the users’ intention to use which in its turn is a determinant of actual use [41]. Since TAM only focuses on individual factors, other models combining individual and situation-specific variables have been developed [42]. For example, the Unified Theory of Acceptance and Use of Technology (UTAUT), developed by Venkatesh et al. [43], combines “perceived usefulness”, “perceived ease of use” with a third determinant for intention to use (i.e., “social influence”) and a direct determinant of actual use (“facilitating conditions”) [43]. Since studies on acceptance and experiences of SFRA have been performed among independently living older adults [44,45], it is pertinent to perform the same type of studies with staff involved in hospital care of older adults.

The overall aim of the study presented in this article was to investigate whether examples of SFRA methods might be relevant for FRA at an orthopedic clinic. The study’s research questions were:

RQ1: In what situations do staff at an acute ward for inpatient orthopedic care assess older patients’ fall risk? Can SFRA support the staff in any of these situations?

RQ2: Which SFRA methods suiting the clinical FRA situations and the clinic’s procedures can be identified from scientific literature?

RQ3: What are the staff’s views on the value of SFRA and its utilization for the clinic?

## 2. Materials and Methods

To address the three research questions, the study adopted a qualitative design with an inductive approach [46] and four sub-studies were performed (Figure 1).

*Sub-study 1* was a focus group aiming at identifying situations where SFRA might assist the FRA currently performed at an orthopedic clinic. Two situations were selected through the focus group interview and written communication between researchers and the clinic.

*Sub-study 2*, published in [47], was a systematic literature review presenting studies where SFRA methods were evaluated in older adults.

*Sub-study 3* was a comparison between the identified SFRA methods (target group, test procedures, and outcomes) from sub-study 2 and the two selected clinical FRA situations from sub-study 1. This sub-study identified SFRA methods which might be relevant for the two selected clinical FRA situations.

*Sub-study 4* investigated the clinical staff’s views of whether and how the SFRA methods identified in sub-study 3 might contribute to their clinical work with FRA in older adults.

### 2.1. Study Setting and Recruitment

Sub-study 1 and 4 were conducted at an orthopedic clinic at a hospital located in a medium-sized Swedish city with approximately 150,000 inhabitants. The clinic performs out-patient and inpatient care (including elective and acute surgical operations), and pre-and post-operative rehabilitation.

For sub-study 1, the participants worked on the acute inpatient ward and were recruited by their managers. They all received written information from their manager and the researchers before providing a written consent to participate.

For sub-study 4, the participants were clinical staff who attended a research seminar held at the orthopedic clinic. They also received oral and written information, through a Mentimeter presentation, prior to providing their written consent.

### 2.2. Data Collection and Data Analysis

#### 2.2.1. Background Information of Participants

The participants were asked to complete study specific questionnaires collecting background information. The quantitative data on the participants’ background information was analyzed by descriptive statistics.

#### 2.2.2. Sub-Study 1: Identification of Clinical FRA-Situations Where SFRA Might Be Relevant

Data on clinical FRA situations where SFRA might be relevant was collected in a focus group interview with clinical staff. The focus group interview aimed to address RQ1.

The interview, which was approximately 90 min, was led by a moderator (first author). Two other researchers (second author and a research engineer) supported the interview in the role of assessors, taking notes and asking complementary questions. All researchers had an engineering perspective, and the authors had experience in collecting user feedback from healthcare staff. To clarify and add information to the interview responses, the participant responsible for the research at the orthopedic clinic also contributed with additional questions to the staff. The interview was audio recorded and transcribed verbatim. In the introduction, the participants were informed that the purpose of the interview was to explore how the staff performed FRA of older patients. As an opening question, they were asked to describe the group of patients aged ≥65 where FRA is relevant. Thereafter, a semi-structured interview guide, containing questions about how fall risk is currently assessed during a patient stay within the ward, and how they thought technology could support their FRA, was used.

The verbatim transcript was analyzed by all three researchers to identify situations where the staff performed FRA of older adults. They identified a list of FRA situations. The brief description of each FRA situation was produced by the moderator who extracted and summarized information on environment, situation, participants, and aim per FRA situation from the assessors’ notes. Thereafter, joint discussions between the researchers on whether or not complementary information from the transcripts should be added were held.

From the identified situations, the researchers selected two FRA situations for further exploration. The selection was performed based on the following criteria: (1) all selected situations should contain FRAs based on motions, i.e., not only questionnaires; and (2) taken together, the selected situations should include different types of FRAs and involve different professions. The FRA situations were selected during a joint discussion between all three researchers until consensus was reached. The results of the analysis were sent to the focus group participants, the staff’s team leader and the clinic’s operational developer for feedback via the person being responsible for research at the clinic. The results were also presented at a staff meeting, in which staff was given the opportunity to provide additional feedback. The feedback was integrated into the brief descriptions of the identified FRA situations.

#### 2.2.3. Sub-Study 2: Systematic Literature Review to Identify SFRA Methods Evaluated with Older Adults

The aim was to identify SFRA methods that had been evaluated with older adults and to contribute with information on the characteristics of each method and the evaluation methodology used. The systematic literature review, previously published in [47], was performed by searching in four databases. The studies were selected systematically according to the set eligibility criteria. Data were extracted using a study specific template with defined variables. The aim of the data analysis was to investigate whether there was evidence of SFRA in terms of discriminative capacity and classification performance, and whether previously identified risk factors for study bias could be identified among the included studies.

The studies’ populations of interest, investigated test results, comparator test results, and outcomes [48] were defined in order to retrieve SFRA methods that might be relevant for the clinical FRA situations identified in sub-study 1.

#### 2.2.4. Sub-Study 3: Identification of Published SFRA Methods Relevant for Clinical FRA

To address RQ2, all studies included in the systematic literature review of SFRA [47], i.e., sub-study 2, were screened to identify articles presenting studies where the SFRA method’s performance in classifying individuals according to their fall risk had been evaluated. The identified articles were subjected to further analysis to identify whether they might be relevant for clinical FRA. The SFRA methods were analyzed according to their perceived relevance for the orthopedic clinic and the two selected clinical FRA situations in sub-study 1. The following criteria were used:(1)The SFRA method used assessment tasks considered to be relevant for performing a FRA in the following situations: walking to the bathroom, the transitions sit-to-stand and stand-to-sit, putting on slippers, walking in stairs, getting into and out of bed, and activities in daily living (ADL)s (Table 1).(2)The SFRA method used one or two sensors since a higher number of sensors was not considered feasible for the clinical setting.

#### 2.2.5. Sub-Study 4: Investigation of the Clinical Staff’s View on SFRA in Clinical FRA

The aim of sub-study 4 was to address RQ3. Data were collected during a research seminar directed towards staff from the entire orthopedic clinic. The seminar included presentations of three research projects of which the overarching study presented in this article was one (Figure 1). The outline of this presentation, which included data collection for sub-study 4, is presented in Figure 2. An online questionnaire, which was accessible via a password-protected webpage (Mentimeter, Stockholm, Sweden/Toronto, ON, Canada), was used to gather the staff’s views on SFRA in clinical FRA.

The presentation (Figure 2) started with a popular scientific background description of sensor technologies used for motion analysis and FRA. Thereafter, the participants received information on the online questionnaire including aim, study procedure, data handling and outline of questionnaire topics and presentation. The volunteering participants opened the online questionnaire at https://www.menti.com (accessed on 1 February 2023) by entering a password.

The first two questions were used to collect written consent to participate and to allow the researchers to use the information provided according to the described data handling. Thereafter, the participants answered six background questions related to their profession, workplace, number of years in the profession, current FRA procedures, use of technology for FRA, and interest in using technology for FRA.

The presentation continued with the provision of brief information on methods used and results obtained in sub-studies 1–3. For sub-study 3, a table presenting the ten selected studies in terms of publication record, FRA method, study population, number of sensors, sensor position(s), and SFRA outcomes was presented. Thereafter, the participants answered ten questions related to each identified SFRA method’s value for the FRA situations and the clinic’s operations.

Thereafter, information on the four SFRA evaluation studies involving patients that were identified in sub-study 2 was provided. Three of these were not selected in sub-study 3 due to not classifying individuals. The participants answered two questions related to the relevance of the FRA methods used in the four studies involving patients. The sequences of questions related to SFRA and FRA methods included two slides explaining the FRA methods. The aim of these slides was to provide information on the methods in case the participants lacked knowledge of them.

For each question, the researchers monitored the number of answers entered in order to allow all participants to answer before proceeding to the next question or information slide. A technical issue with Mentimeter was that also participants entering the code after the written consent questions had been asked were able to answer all remaining questions, i.e., without providing consent to participate and permitting the researchers to use the provided information. Therefore, the answers from these people were omitted in the analysis. The lack of a possibility to provide written consent in later stages of the questionnaire is a limitation in the Mentimeter questionnaire tool.

The analysis of the data depended on the nature of the question. Multiple-choice questions were analyzed by summarizing the number of answers per response alternative. Free-text questions were analyzed by coding the content of each answer, clustering codes according to similarity, and summarizing the number of answers per cluster. The number of responses per question was also counted. For certain clusters, quotes illustrating the participants’ views were also extracted. The first author made a first analysis which was reviewed by the second author. Differences in views on how to interpret responses were discussed until consensus was reached.

## 3. Results

The study addressed the following three research questions:

RQ1: In what situations do staff at an acute ward for inpatient orthopedic care assess older patients’ fall risk? Can SFRA support the staff in any of these situations?

RQ2: Which SFRA methods suiting the clinical FRA situations and the clinic’s procedures can be identified from scientific literature?

RQ3: What are the staff’s views on the value of SFRA and its utilization for the clinic?

### 3.1. Sub-Study 1: Identified Clinical FRA-Situations Where SFRA Might Be Relevant

Five individuals participated in the focus group interview. One was responsible for the research at the clinic and the others represented different categories of health personnel including assistant nurse, nurse, occupational therapist, and physiotherapist. The assistant nurse and the nurse worked on an acute inpatient ward while the physiotherapist and the occupational therapist worked both in inpatient and outpatient care. As for experience in the field, three individuals had <5 years working with patients ≥65 years of age, and one had 5–10 years of experience working in a similar setting. All four reported assessing fall risk in older adults several times per day and voiced an interest to use technology in FRA. They further expressed that the technology could contribute to fall prevention as well as to recognize some risks.

Six clinical FRA situations were identified from the focus group interview (Table 1). The situations ‘in-ward mobility training and activities in daily living (ADL)’ and ‘preparations (assessment and training) for discharge from hospital’ were selected for the subsequent work in sub-study 3 and sub-study 4.

### 3.2. Sub-Study 2: Identified Published SFRA Methods Evaluated with Older Adults

Based on results published in 33 articles, the systematic review of SFRA literature, previously published in [47], identified evidence of that SFRA can be effective in assessing older adults’ fall risk, both in terms of discriminating groups with varied levels of fall risk from each other and in terms of classifying individuals based on fall risk. Worth noticing is that only 4/33 of the included articles involved patients as study population in the evaluations: Caby et al. [50] and Marschollek et al. [51] involved inpatients at geriatric clinics; Joseph et al. [52] involved older patients hospitalized at a trauma clinic after falling; and Genovese et al. [53] involved older patients from clinical partners in a European collaboration project.

### 3.3. Sub-Study 3: Identified Published SFRA Methods of Relevance for Clinical FRA

Fifteen out of the 33 articles in sub-study 2 [47] evaluated the SFRA method’s performance in classifying individuals based on fall risk. Ten of these were considered relevant for the orthopedic clinic and the two selected clinical FRA situations from sub-study 1. A flow diagram of the analysis is presented in Figure 3.

The ten selected articles are presented in Table 2. Only one article [51] involved inpatients in a geriatric clinic while the others involved community-dwelling older adults (7 articles), a convenience sample (1 article) and a sample recruited from several sources (1 article). Six of the articles monitored specified tasks (walking, Timed Up and Go (TUG) test or standing balance) and four used daily life measurements (walking, identified sit-to-walk and walk-to-sit transitions, or activities). Most articles (7/10) used only one sensor which was positioned on the lower back/pelvis in six articles and on the wrist in one article. The three articles that used two sensors positioned those on the shins/shanks (2 articles) or upper and lower trunk (1 article). The most frequently used sensor type was 3D accelerometers (9/10), either alone (6/10) or in combination with other sensor types (3/10) including a 3D gyroscope and a heart rate monitor. Moreover, one article used only a 3D gyroscope. In most articles (8/10), the SFRA outcome was classifying the older adult as a “non-faller” or a “faller”. However, three classes (“non-faller”, “faller”, “multiple-faller”) were used as SFRA outcomes in two articles.

### 3.4. Sub-Study 4: The Clinical Staff’s View of SFRA in Clinical FRA

The study included 13 participants. As shown in Table 3, the group represented several categories of health personnel such as assistant nurse, nurses, physicians, and physiotherapists working both with in- and outpatient care. The majority of the participants had worked >10 years in their profession and performed FRA in their clinical work. They used various FRA methods, where observation was the most common. Most participants never used technologies in FRA but expressed a varying degree of interest in using technology in FRA.

The participants described that the patients’ fall risk is assessed in various settings some of which included meetings with outpatients, as well as during inpatient enrollment, post-surgery rehabilitation planning, etc. They described that they observe and collect information both from the patients and their caregivers/relatives (Table A1). They meet a broad range of patients, e.g., patients with injuries caused by falls or other reasons, and patients with a varying level of balance and walking abilities. Most participants also meet patients with arm injuries inhibiting arm movements (Table A1).

The remainder of this section presents the participants’ views on SFRA in terms of: relevance and feasibility of the assessment tasks (Section 3.4.1, the type of information desired from the assessments (Section 3.4.2), the participants’ willingness to dedicate time on related work (Section 3.4.3), envisioned barriers to using SFRA in clinical work (Section 3.4.4), and anticipated outcomes from using it in clinical work (Section 3.4.5).

#### 3.4.1. Views on Relevance and Feasibility of Assessment Tasks Used in SFRA

Some participants were familiar with ≥1 of the FRA methods used in the SFRA approach selected in sub-study 3. As shown in Figure 4, these FRA methods included both defined tests (TUG, walking and standing balance tests) and gait and activities in daily life. Activities in daily life was more well-known among the participants than the other FRA methods.

The FRA methods found to be most relevant in clinical work were activities in daily life followed by gait in daily life and gait tests. Interestingly, only one respondent perceived that TUG was relevant in clinical work. Moreover, all FRA methods were found feasible for the clinical work of 3–5 participants.

Sub-study 2 identified three evaluations of SFRA methods’ abilities to discriminate between groups of patients with different fall risk levels [50,52,53]. Despite not being considered as relevant for clinical FRA according to the criteria in sub-study 3, the researchers were interested in whether the participants found these FRA methods potentially relevant, i.e., elbow flexion test with the patient performing as many cycles of elbow flexion–extension as possible during 20 s in bedbound position [52] and 6-Minutes Walking Test (6MWT) with the patient walking back and forth on a 30-m walkway for 6 min [53]. No one was familiar with the elbow flexion test and only one participant was familiar with the 6MWT. However, after being introduced to the concepts of the two FRA methods, some participants reported that they could be relevant for their clinical work (Figure 5).

#### 3.4.2. Type of Information Desired from SFRA

The participants stated that they would like to obtain information on their patients’ fall risk, both in general but also in specific activities/situations from SFRA (Figure 6).

#### 3.4.3. Willingness to Dedicate Time to SFRA

The participants’ willingness to dedicate time to SFRA-related work, i.e., to mount and remove sensors, and to review SFRA results, varied (Figure 7). The maximum amount of time reported was 30 min, although one of the participants providing this answer stated that the amount of time depended on the patient’s needs. On the other hand, some participants were not willing to dedicate time to SFRA. Moreover, some participants stated that they did not know how much time they were willing to dedicate to SFRA, and one of them questioned if SFRA should be performed in outpatient care where the scheduled visits are short.

#### 3.4.4. Envisioned Barriers to Using SFRA in Clinical Work

Four out of eight participants envisioned time constraints as a barrier for using SFRA in their clinical work (see Table A1). One of them stated “unless the system works smoothly and is easy to use”. Lack of resources and equipment were also envisioned as barriers for clinical use. Other participants raised questions on whether the patients want SFRA and what it would lead to.

#### 3.4.5. Anticipated Potential Outcomes from Using SFRA in Clinical Work

The participants anticipated both positive and negative outcomes from using SFRA in their clinical setting (see Table A1). A major potential value of SFRA is fall prevention and subsequent reduction in injuries, related surgery, and hospital visits. In addition, SFRA can increase objectivity in FRA for identifying risk patients, raise awareness, and increase patient security and safety. An additional foreseen potential value of SFRA includes support in resource management and focusing efforts. Negative outcomes anticipated included time constraints, cost, waste of resources, as well as insufficient efficacy, and reliability.

## 4. Discussion

This study aimed to investigate whether examples of SFRA methods might be relevant for FRA in an orthopedic clinic. The study addressed the three research questions:

RQ1: In what situations do staff at an acute ward for inpatient orthopedic care assess older patients’ fall risk? Can SFRA support the staff in any of these situations?

RQ2: Which SFRA methods suiting the clinical FRA situations and the clinic’s procedures can be identified from scientific literature?

RQ3: What are the staff’s views on the value of SFRA and its utilization for the clinic?

The major results of the study in relation to RQ1–RQ3 are the following:

RQ1: In the acute inpatient orthopedic ward, FRA is performed in several situations during a hospital stay. The FRA is mainly performed through the staff’s observations while structured FRA is performed during patient enrollment and post-surgery. The patient’s ability to safely perform daily activities and move is at focus. Several professions (assistant nurses, nurses, occupational therapists, physicians, and physiotherapists) contribute to clinical FRA.

Although Cameron et al. [19] identified that multifactorial interventions may reduce rate of falls in hospitals in their review, they could not draw conclusions on the effectiveness of specific intervention elements such as FRA. However, the FRA situations identified in the study presented in this article are in accordance with methods and guidelines described for Swedish caregivers and healthcare personnel in “The Handbook for Healthcare” [63], which describes that FRA within in-hospital emergency service should be performed when the patient arrives to the hospital, pre- and post-surgery, at states of deteriorating health, confusion, or anxiety. They also describe that the staff should offer the patients physical activities and consult occupational therapists and physiotherapists for ADL- assessment and training, as well as balance and muscle strength training. The Handbook for Healthcare advocates the use of a clinical FRA instrument on the arrival to the hospital emergency service.

RQ2: There is evidence in scientific literature that SFRA can discriminate between groups of older adults with different levels of fall risk and classify older adults according to risk levels. However, many SFRA evaluations involve older adults living independently and base the FRA on clinical tests (gait, balance, etc.) performed under observation. Nevertheless, it was possible to identify ten examples of evaluated SFRA methods which classified older adults according to fall risk, used 1–2 sensors, and included assessment tasks reflecting activities and movements of daily life (walking, transitions sit-to-stand and stand-to-sit, standing on one and both legs, etc.). These examples support Shany et al.’s [32] conclusion that SFRA can assist FRA in hospital wards by collecting data in semi-supervised or supervised FRA settings including only one or two assessment tasks.

RQ3: In the entire orthopedic clinic (including both in- and outpatient care), FRA is mainly performed by observations and the staff’s clinical experience. In addition, medical records, patients’ own descriptions, patient characteristics, physiological measurements, and structured questionnaires are used as complementary sources of information. FRA is mostly performed without technology and several healthcare professions contribute.

The staff working with in- and outpatient orthopedic care find that SFRA tasks reflecting activities and movements of daily life are relevant and feasible for their clinical work. More specifically, FRA based on gait tests, and gait and ADL were found clinically relevant by a larger number of staff compared with the TUG and standing balance tests. Despite not being familiar with the elbow flexion test, some staff perceived that the test might be relevant for FRA after a brief introduction to the concept.

The staff wanted SFRA to contribute with information on a patient’s fall risk, both in general and in specific situations or activities. Context-specific fall risk information was also found being highly useful by older adults wanting to better understand their fall risk [45]. Some staff are willing to dedicate 15–30 min/day on SFRA-related work (i.e., mounting- and removing sensors, and reviewing SFRA results). Some staff are not willing to dedicate any time, and some do not know. It was stated that the amount of time depends on the patient’s need. It was also questioned whether the SFRA should be performed during a short patient–physician meeting at the clinic or somewhere else. However, for example Shany et al. [32] reported that lack of time or equipment can hinder staff working in busy clinical settings from performing thorough or highly objective FRA. The staff identified time constraints as a major barrier for using SFRA in clinical work. It was emphasized that SFRA systems need be easy-to-use, work smoothly, and be reliable to prevent stress among the staff. System reliability has also been identified as important for older adults’ adherence to self-assessment of fall risk [44]. These views are in accordance with UTAUT [43] which describes that “perceived ease of use” (reflected in this study by requirements on “easy-to-use” and “work smoothly”) has an impact on “intention to use” which is a determinant of “actual use”.

The staff expressed concerns (such as questioning whether the patients would accept the technology and anticipated non-effective SFRA as costly and a waste of resources). They also expressed potential values (such as raised awareness, focused efforts on risk patients increasing patient security and safety, and support to resource management) related to SFRA. Both the concerns and potential values may be related to the perceived usefulness of SFRA, which according to UTAUT [43] is a determinant to technology acceptance and usage. Moreover, their emphasis that SFRA needs to be implemented in ways and contexts that effectively prevent falls and thereby reduce the number of fall-related surgeries and patient visits might illustrate the importance of facilitating conditions, which is a direct determinant to actual use according to UTAUT [43].

## 5. Conclusions

Although SFRA has mainly been evaluated among older adults living independently and by collecting sensor data from assessment tasks performed under observation, this study has shown that SFRA includes FRA methods that may be relevant in orthopedic clinics. In this study, clinical staff from in- and outpatient orthopedics care expressed that, in order to be clinically relevant, SFRA must be effective in reducing falls, reliable, smooth, and easy-to-use. These views are supported by the UTAUT model.

## Figures and Tables

**Figure 1 sensors-23-01904-f001:**
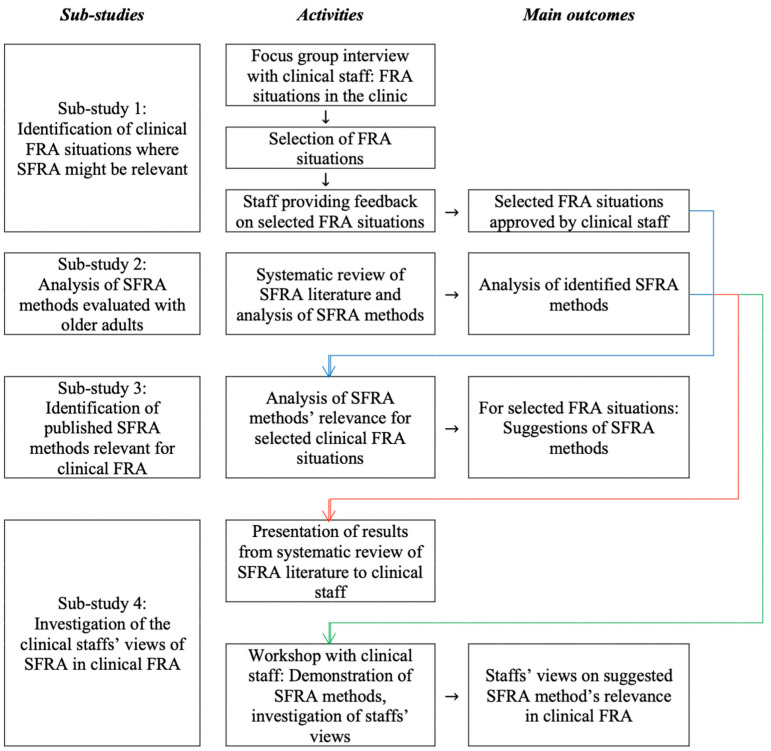
Overview of the study design including the four sub-studies, their related research activities, and main outcomes.

**Figure 2 sensors-23-01904-f002:**
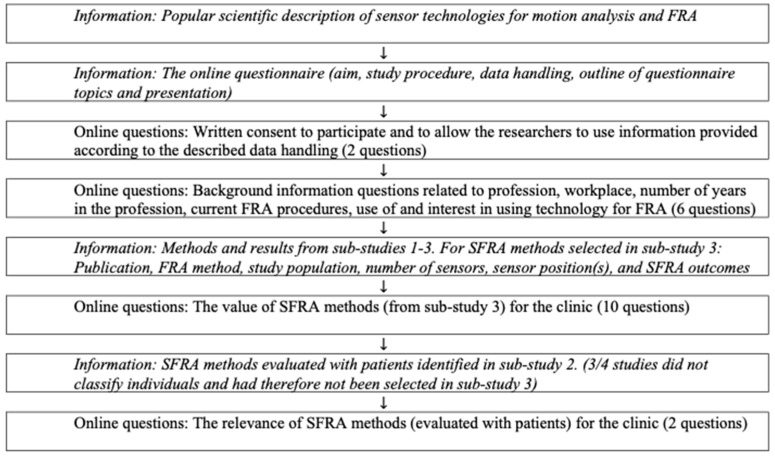
Outline of the presentation where the online questions constitute sub-study 4.

**Figure 3 sensors-23-01904-f003:**
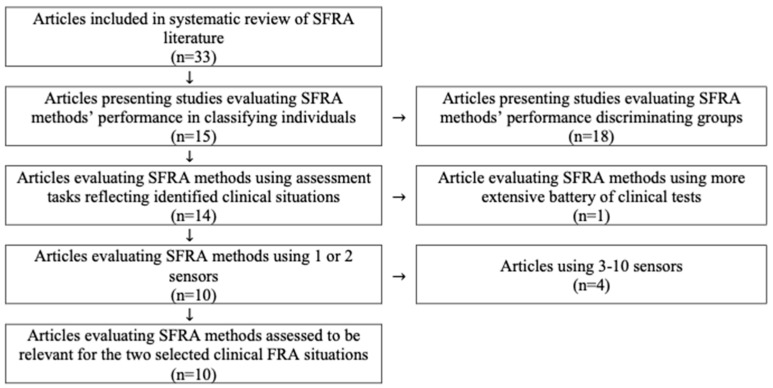
Flow diagram of articles analyzed for relevance for the selected clinical FRA situations from sub-study 1.

**Figure 4 sensors-23-01904-f004:**
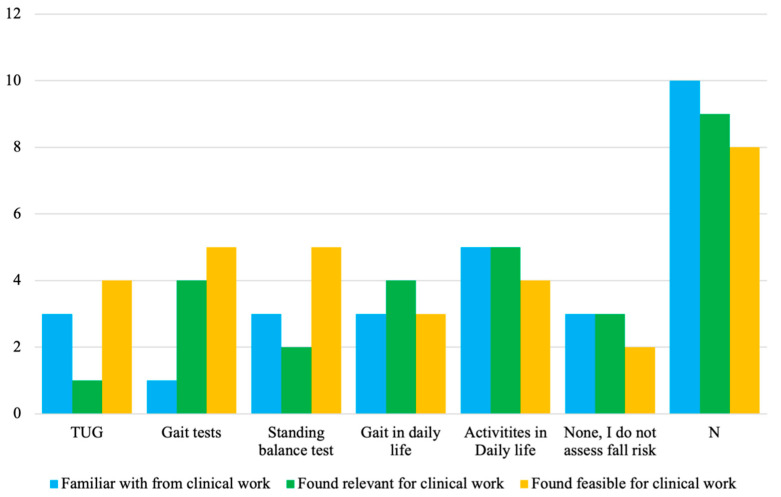
The participants’ familiarity with, and the perceived relevance and feasibility of, the FRA methods used in the SFRA methods selected in sub-study 3.

**Figure 5 sensors-23-01904-f005:**
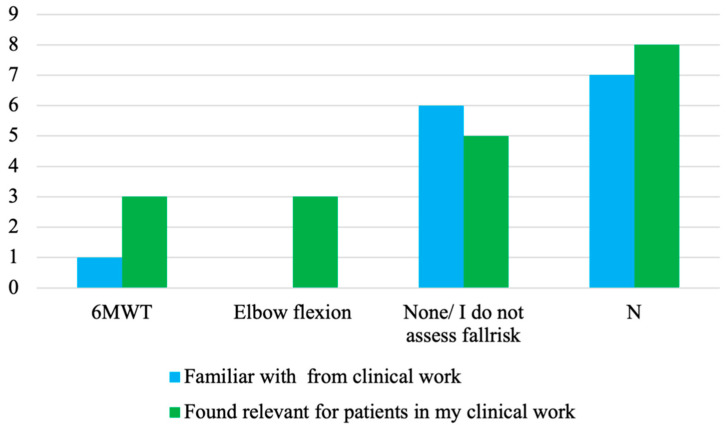
The participants’ familiarity with, and perceived relevance of, the two FRA methods used in the SFRA studies with patients that were not selected in sub-study 3.

**Figure 6 sensors-23-01904-f006:**
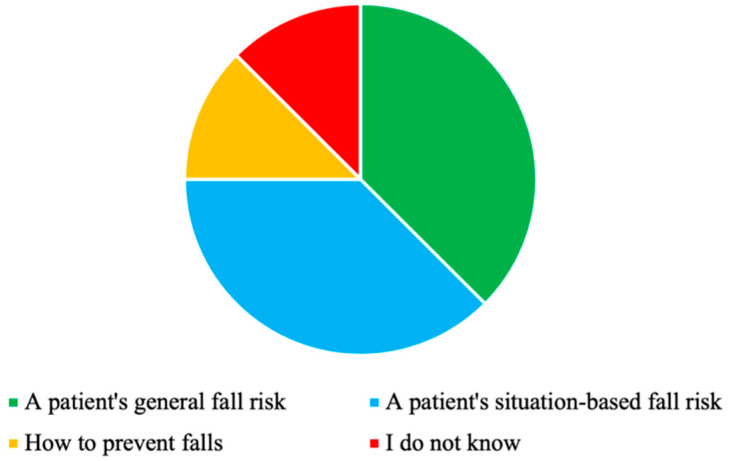
Information that the participants would like to obtain from SFRA (N = 8).

**Figure 7 sensors-23-01904-f007:**
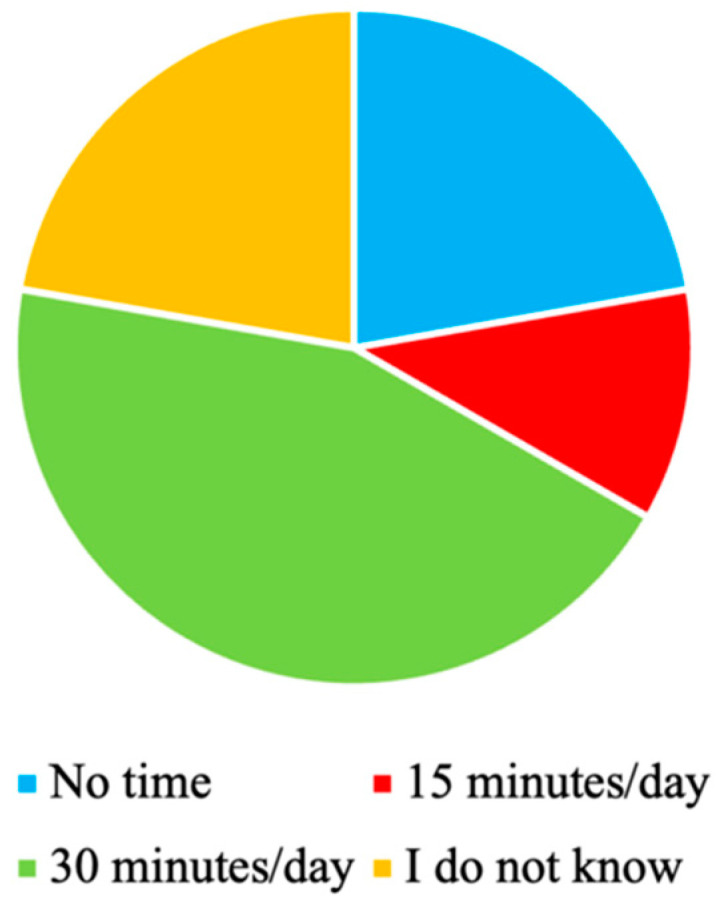
Maximum time that the participants are willing to dedicate to SFRA (N = 8).

**Table 1 sensors-23-01904-t001:** Clinical FRA situations identified in a focus group interview with clinical staff. The selected FRA situations to explore further regarding SFRA are highlighted in italics.

FRA Situation	Aim	Context	Assessor	Method
Trauma patient arrives to ward prior to surgery	To investigate the patient’s needs of fall preventive measures, e.g., bed gate’s height, non-slip socks, motion alarm, assistive technology in the establishment of a care plan. Documented in medical record.	In the patient’s bed	Nurse and assistant nurse, consulting physician in case of questions. Can also be performed by physician.	Part of structured, comprehensive risk assessment and measures using the clinical FRA instrument Downtown Fall Risk Index (DFRI) [49].
After surgery (same day)	To investigate whether the fall preventive measures taken, and care plan need to be modified. Documented in medical record.	In the patient’s bed	Nurse and assistant nurse, consulting physician in case of questions. Can also be performed by physician.	Part of structured, comprehensive risk assessment and measures using the clinical FRA instrument DFRI [49] and based on general health conditions and physiological measurements (blood pressure, pulse, temperature, and respiratory rate).
First mobilization (day 1 post-surgery)	To assess the risk of falling at the ward and mobility (in planning of mobility training and provision of assistive technology).	In the patient’s room, initially on the edge of the bed	Physiotherapist and/or occupational therapist. Assistant nurse is also often present.	Evaluation of the patient’s mobility by a stepwise test using tasks with increased levels of difficulty: (i) Sitting on bed/chair; (ii) Sitting steadily on bed/chair; (iii) Standing; (iv) Standing and lifting one foot; (v) Walking; (vi) Walking a little longer. Observation and assessment of patient mobility, documented as start notes. Initial tests performed with a standard walker. The patient is often in pain. Patients with hip fractures are initially unable to stand on their legs due to pain, fear, or dizziness. Patients with upper arm fractures practice managing walking aids with one arm.
*In-ward mobility training and activities in daily living (ADL)*	To assess mobility and risk of falling and use to adjust need of help and training efforts in the ward.	In the patient’s room, initially on the edge of the bed	Nurse and assistant nurse, on some occasions also physiotherapist/occupational therapist.	Observations of the patient’s mobility in training and ability to perform ADLs (e.g., going to the bathroom, sitting down/standing up from seated on toilet, putting on slippers). No standardized assessment or documentation of fall risk performed.
Hip rehabilitation training (6–12 weeks post-surgery)	Structured training focused on rehabilitation after hip prosthesis surgery (offered to trauma patients who were previously very active).	In a specific open area in the hospital	Physiotherapist and assistant nurse.	Circle training in group with stations for balance- and resistance exercises. Led by physiotherapist and assistant nurse from the hospital’s rehabilitation unit; 45–60 min once a week for 6 weeks starting 6 weeks post-surgery.
*Preparations (assessment and training) for discharge from hospital*	For patients living in the community: To assess several ADLs that are needed to be able to manage living independently. Identification and recommendations for adjustments and municipal rehabilitation. For patients in special housing: Transfer to special housing unit.	In the patient’s room, initially on the edge of the bed and successively including ADLs occurring at home	Nurse and assistant nurse, on some occasions also physiotherapist/occupational therapist.	Training of certain ADLs that the patient needs to perform at home (stairs, in and out of bed, bathroom visits, etc.), discussions about needs for municipal rehabilitation.

**Table 2 sensors-23-01904-t002:** SFRA methods identified in [47] and considered to be relevant for the orthopedic clinic and the two selected clinical FRA situations from sub-study 1. n-f: non-faller; f: faller; m-f: multiple-faller; Accel: 3D accelerometer; Gyro: 3D gyroscope; TUG: Timed Up and Go.

Study, Year	Ref No	Assessment Task	Study Population	Number of Sensors	Sensor Types	Sensor Positions	SFRA Outcomes
Marschollek, 2011	[51]	TUG test and walking (20 m)	Inpatients (geriatric)	1	Accel	Lower back	n-f/f
Bautmans, 2011	[54]	Walking (2 × 18 m)	Several sources for recruitment	1	Accel	Pelvis	n-f/f
Doi, 2013	[55]	Walking (15 m)	Community-dwelling	2	Accel	Upper and lower trunk	n-f/f
Greene, 2014	[56]	TUG test	Community-dwelling	2	Accel + Gyro	Shin/shank	n-f/f
Ihlen, 2016	[57]	Walking (daily life)	Community-dwelling	1	Accel	Lower back	n-f/f (f ≥ 2 falls)
Ihlen, 2016	[58]	Walking (daily life)	Community-dwelling	1	Accel	Lower back	n-f/f (f ≥ 2 falls)
Iluz, 2016	[59]	Identified sit-to-walk and walk-to-sit transitions in daily life	Convenience sample	1	Accel	Lower back	n-f/f (f ≥ 2 falls)
Greene, 2017	[60]	TUG test	Community-dwelling	2	Accel + Gyro	Shanks	n-f/f
Ghahramani, 2019	[61]	Standing balance tests	Community-dwelling	1	Gyro	Lower back	n-f/f/m-f
Yang, 2019	[62]	Activities in daily life	Community-dwelling	1	Accel and heart rate	Wrist	Three classes: n-f/f/m-f Two classes: (n-f + f)/m-f or n-f/(f + m-f)

**Table 3 sensors-23-01904-t003:** Background characteristics of participants in sub-study 4. For each question, the total number of responses (N), the response alternatives, and the number of responses per response alternative (n) are presented.

Question Asked to Participants (N)	Response Alternatives	n
What is your health profession? (N = 12)	Assistant Nurse	1
	Nurse	5
	Occupational Therapist	0
	Physician	3
	Physiotherapist	2
	Other	1
For how long have you worked in your profession? (N = 10)	>10 years	9
	5–10 years	0
	<10 years	1
Where in the orthopedic clinic do you work? (N = 12) *Possible to select several response alternatives.*	Acute inpatient ward	3
	Elective inpatient ward	1
	Orthopedic outpatient care	6
	Rehabilitation outpatient care	1
How do you assess fall risk of patients ≥65 years of age today? (N = 12) *Free-text question, possible to provide more than one answer.*	Observations/intuition	10
	Medical record	3
	Patient’s own descriptions	2
	Patient characteristics (pharmaceuticals, age, etc.)	2
	Physiotherapist’s assessment	1
	Norton Score (including fall risk)	2
	Physiological measurements	1
Do you use technology in FRA? (N = 13)	No, never	10
	Yes, seldom	1
	Yes, often	2
How interested are you in using technology in clinical FRA? (N = 13) *One respondent was both very and extremely interested.*	Extremely interested	1
	Very interested	3
	Moderately interested	4
	Slightly interested	4
	Not at all interested	2

## Data Availability

Not applicable.

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
