# Peer review of "Clinical Sensor-Based Fall Risk Assessment at an Orthopedic Clinic: A Case Study of the Staff’s Views on Utility and Effectiveness"

_sensors, 2023, doi:10.3390/s23041904_

Round 1
Reviewer 1 Report
Thank you for your hard work on the manuscript. I have made some specific comments on the English language and style. Also, would suggest to review and adjust your abbreviations for appropriateness. Some sections are very hard to follow and repetitive.

Author Response
RESPONSE TO REVIEWERS
Dear Editor and Reviewers,
Thank you for your valuable comments on the manuscript “Clinical Sensor-Based Fall Risk Assessment at an Orthopedic Clinic: A Case Study of the Staff’s views on Utility and Effectiveness” (revised title)
Your comments have resulted in several improvements. Below follows point-by-point information on how the comments from each reviewer have been addressed in this major revision of the manuscript. In addition, we have checked the English language and modified the layout of Figure 6 to make it more in similar toFigure 7. Changes made within the manuscript are indicated by track changes.
Please note that the numbering of lines in the edited manuscript is incorrect from page 2 in the document. The error occurred in the editing of the text, unfortunately we have not been able to correct it.
Reviewer 1
- General Comments
Reviewer’s comment: Appreciate author’s hard work on this manuscript. This is the first time I have reviewed a manuscript for this specific journal (Sensors). Therefore, I had to review the manuscript submission guideline (Authors Guidelines) before I could prepare my review and provide comments. I may make some comments on my review that may be acceptable for the Sensors Editorial Board and I have misread the notes. With regards the English Language (vocabulary and grammar), I have made comments about terms to be more professional sounding. The research has been completed, so there are some areas with is vs. was is required, plural vs. singular. One additional comment is some very unusual abbreviations. For example, I have never seen mentor to be abbreviated as (M), nurse is RN vs. N, physician is Dr. or MD and not P, finally other researcher (AK). This got very confusing during the review. I do understand the authors have provided a list of abbreviations, but for overall readership, this should be noted and corrected.
Authors response: We thank the reviewer for the constructive feedback. Descriptions on how we have addressed the comments related to English language and abbreviations are found below.
- Specific Comments
TITLE:
Reviewer’s comment:
I would suggest revising the title, sounds very odd reading it. It sounds like if you reverse the title, it will be better. Is this a case study or an overall review. Here is my opinion if you wish to accept. Clinical Sensor-Based Fall Risk Assessment at an Orthopedic Clinic: A review of Clinical Staffs’
Vies on utility and effectiveness
Authors response:
The title has been revised to “Clinical Sensor-Based Fall Risk Assessment at an Orthopedic Clinic: A Case Study of the Staff’s views on Utility and Effectiveness”
ABSTRACT:
Reviewer’s comments:
Line #8 – change “is” to “are”, since we are discussing ‘falls” (multiple/recurrent). Line #8 – I think “severe” doe not fit well here, consider changing to “serious”.
Line #10 – delete “clinically’ since you have already mentioned its clinical use, just saying “identify meaningful SFRA methods… is appropriate and delivers the concept.
Line #19 – Change “hinder” to “barrier”, better terminology.
Authors response:
All suggested changes have been implemented. We also made some other clarifying changes in the abstract.
INTRODUCTION:
Reviewer’s comments:
Line #26 – Change ‘is” to “are” as above; if possible, consider rewording to: Falls are major threat to the health of older adults resulting in injuries and even premature death.
Line #28 – Consider using “≥ “for adults ≥ 65 years of age (can apply to other sections).
Line #29 – Delete Patients falling, just say “moreover, falls are the most common adverse events reported in hospital with a rate ranging from 1.3 – 21 falls /patient/day
Line #32 – Not sure if the word “accident” adds to the content. Just delete and say “10% of all falls occurring among older adults cause severe injuries, most commonly fractures”.
Line #32 – Line #37 (second paragraph) – Repetitive wordings.
Line #49 – Change “is” to “are”.
Line #62 – Consider a minor revision to avoid extra wordings, “there is a major need for tools/apps that are easy-to use, generate objective, accurate, and quantitative risk assessment in clinical settings.
Line # 63 – Line #64 – This has generated an interest in sensor-based FRA…..
Line #65 – Line #66 – Delete the second SFRA and just say “have identified a large variation in these tools measured parameters, assessment tasks, …
Line #69 – Here you use UX for user experience and not sure if this is an appropriate abbreviation, why not just spell in out…
Line #71 – Line #72 – Here it should just say the concept of User Experience is to focus on the design and the predication of its utilization (technology acceptance).
Line #77 – Here “AU”, as actual use not sue why abbreviated!! Going down this paragraph, the abbreviations just caused fatigue!!!!
Line #87 – Just say “in what situations do staff at an acute…..Delete “working”.
Line #91 – Consider revising to “What are the staff’s views on the value of SFRA and its utilization for the clinic?
Authors response:
All suggested changes have been implemented and Line #32 – Line #37 (second paragraph) have been rephrased. However, a different phrasing than the one suggested on Line #29 was used since an important detail was missing. The statistics reported were to reflect the number of falls during 1,000 patient days, not per day.
Regarding the comment on line #77, all abbreviations for the TAM- and UTAUT-constructs have been removed and the paragraph has been reformulated
MATERIALS AND METHODS
Reviewer’s comments:
Line #99 – Delete the second “study” already have it at the beginning.
Line #100 – Line #102 – Limit the wording and say: Two situations were identified and selected between the focus group and the focus group (researchers) and the clinic.
Line #104 – Change “with” to “in older adults”.
Line #111 – On line #104 you used older adults” you may want to keep this consistent and change this to older adults as well.
Line #114 – You have listed an approximate 150,000 inhabitants, is this/year or per month. Not sure if “inhabitants” is a good term, may just use simple “patients” or “population” or “residents”.
Line #117 – Line #122 – Here is some word eliminations. For sub study 1, the participants worked on the acute in-patient care ward and were recruited by the manger; they all received written and verbal information form their manager as well as the researchers before providing informed consent.
Line #124 – Say They also received written and verbal information through a ……and end at the consent.
Line #129 – Delete the “in sub-study 1 and 4’ it is in the title. Just start the sentence with: the participants were required to provide specific background information via a questionnaire and this information was analyzed by descriptive statistics.
Line #143 – you are repeating the titles which is not necessary. Start with “an interview process with clinical staff addressed the RQI (…..)
The information between Lines #138 – Line #151 (two paragraphs) can be aligned and eliminate unnecessary wording. Here is my suggestion if you wish to consider: Line #138 – Line #139 – Not sure if you need to report where the interview took place it is not relevant. But, may just start with: The interview which was approximately 90 minutes, was led by a moderator please delete the (ME) abbreviation!!! Two other researchers (Ak and a research engineer) …. Consider adding “AK” to your abbreviation list, did not see this listed. Now consider all the other information in these two paragraphs.
Line #142 – Use ≥ 65
Line #146 – Delete (ME) and just spell out moderator.
Line #152 – Line #162 - Here again lots of redundant section, not sure if you need to report “verbatim” transcript; maybe you should just say “The collected data was analyzed by all three researchers for the purpose of extracting situations where the staff performed FRA on older adults. They identified list of
Also, not sure if “scenarios” is a better term than “situations”.
Line #167 – Delete “which contained brief description” it is repetitive, just say the result of the analysis was submitted to the leaders of the focus group and further presented in staff meetings and staff was given the opportunity to provide additional feedback.
Line #176 – Line #179 – I think you should be able to delete this first three lines and just start the paragraph with The studies population of interest…..
Line #200 – Line #203 - Again, delete the parenthesis – readers can refer to the text. The data was collected during a research seminar and included presentations from all three research projects.
Line #249 – Please spell out mentor and as mentioned earlier, list AK on the abbreviation list.
Authors response:
All suggested changes have been implemented except for the following:
Line #114: Here we mean the number of citizens in the city and have changed to: “…in a medium-sized Swedish city with approximately 150,000 inhabitants”.
Line #117 – Line #122: changed to “For sub-study 1, the participants worked on the acute inpatient care and were recruited by their managers. They all received written information from their manager and the researchers before providing a written consent to participate.”
Lines #138 – Line #151: The abbreviations of author names have been exchanged to first and second author
Line #167 has been changed to “The results of the analysis were sent to the focus group participants, the staff’s team leader and the clinic’s operational developer for feedback via the person being responsible for research at the clinic. The results were also presented at a staff meeting, in which staff was given the opportunity to provide additional feedback.”
Line #176 : The first sentence was kept.
Line #200 – Line #203: The research seminar contained presentation of from three different research projects of which one where the three sub-studies. We have rephrased this sentence to clarify this.
Line #249: We used first and second author instead.
We decided to not switch from situation to scenario.
RESULTS:
Reviewer’s comments:
Line # 261 – Change ‘people” to “individuals” more professional terminology.
Line#261 – Line #262 - Just say one responsible for the research at the clinic and others representing different disciplines like PT, OT, nurse (RN), and AN. The RN and AN worked in in-patient orthopedic ward while the PR and OT worked both in …
Line #265 – Line #266 – As for experience in the field, three individuals had <5 years working with subjects ≥ 65 years of age and one had 5-19 years of experience working in a similar setting.
Line #267 – change to all four reported assessing fall risk in older adults several times per day and voiced an interest in the technology use. They further expressed that the technology could contribute to fall prevention as well as recognize some risks.
Line #247 – Preparations does not need to be capitalized…
Line #280 – Should this be specified that the results were published in 33 articles, I am not clear!! Line #284 – EU project??? What is this?
Line #287 – Revise to: …in sub-study 2 evaluated the SFRA ….
Line #294 – Here you say only one article involved patients (what patients?) and then you go on with the others involved community dwelling older adults!!!
Comment: We have specified that the one article involved inpatients at a geriatric clinic
Line #314 – I think I have seen physicians being abbreviated as (MD’s)…
Line #316 – Delete the “.” Between profession and performed.
Line #318 – Maybe adding varying degree of interest???
Line #324 – Line #332 – The term “participants” has been used multiple times. Start with: the participants described that the patients fall risk was assessed in various settings some of which included out-patients, during enrollment, post-surgery rehabilitation planning, etc. The observation and information were collected both from the patients as well as their caregiver/relatives.
Line #331 – I would suggest making this a separate sentence. Patients with upper extremity injuries (i.e., arms) were also identified and evaluated.
Line #333 – Line #337 – You have repeated SFRA in each line here. Combine the contents, the type of information, the willingness to dedicate time, the barriers, and the consequences of ….At least two can be eliminated.
Line #340 – delete the second “methods”. Maybe “approach” is a better term as well. “FRA approach”.
Line #351 – Delete “SFRA Methods” and say SFRA evaluations and its abilities to …..
Line 363 – Change “wanted” to “desired” or “required” more professional and better suited, because the participants were requesting them. In addition, should this also emphasize “more valuable information”??
Line #369 – Consider change to: “Willingness to dedicate/commit time to SFRA”.
Line #370 – Start the sentence with: “Per participants, this was defined as the time spend to mount……
Line #375 – What did they not know??? Also, may consider changing the last part of this sentence to: one of them also suggested that SFRA may be a better approach in out-patient setting based on the limited time if the scheduled visits.
Line #380 – Change “hinders” to “barriers”: Envisioned Barriers of SFRA in Clinical Setting”
Line #384 – Same as above, “barriers”.
Line #387 – Change “foresee” to “anticipated” and “consequences” to “outcomes” or “values” I think fits better her for the content. The participants anticipated both positive and negative values based on the utility of SFRA in their clinical settings.
Line #388 – Here is a revision suggestion for this sentence: A major value would be fall prevention and subsequent reduction in injuries, related surgeries, and hospital visits.
Line #390 – Just start with in addition, SFRA could increase the objectivity of FRA identifying at risk patients, raise awareness, and increase patient safety and security.
Line #393 – Line #395 – Just combine and say: the negative values perceived included: time constraints, cost, efficacy, reliability, and waste of resources.
Authors response:
The following changes have been implemented according to the reviewer’s comments:
Line # 261 – Change ‘people” to “individuals” more professional terminology.
Line #247 – Preparations does not need to be capitalized…
Line #287 – Revise to: …in sub-study 2 evaluated the SFRA ….
Line #294 – Here you say only one article involved patients (what patients?) and then you go on with the others involved community dwelling older adults!!!
Line #316 – Delete the “.” Between profession and performed.
Line #318 – Maybe adding varying degree of interest???
Line #333 – Line #337 – You have repeated SFRA in each line here. Combine the contents, the type of information, the willingness to dedicate time, the barriers, and the consequences of ….At least two can be eliminated.
Line 363 – Change “wanted” to “desired” or “required” more professional and better suited, because the participants were requesting them. In addition, should this also emphasize “more valuable information”??
Line #369 – Consider change to: “Willingness to dedicate/commit time to SFRA”.
Line #380 – Change “hinders” to “barriers”: Envisioned Barriers of SFRA in Clinical Setting”
Line #384 – Same as above, “barriers”.
Line #393 – Line #395 – Just combine and say: the negative values perceived included: time constraints, cost, efficacy, reliability, and waste of resources.
The following comments from the reviewer has been addressed as follows:
Line#261 – Line #262 We have removed all staff abbreviations and used the term “categories of Health personnel” (MESH-term) instead of discipline. We have also changed in-patients to inpatient (MESH-term), and out-patient to outpatient (MESH-term) and kept the word acute in “an acute inpatient ward”.
Line #265 – Line #266 We used “patients” instead of “subjects”
Line #267 – we changed the suggestion slightly (to “…interest to use technology in FRA…” instead of “…interest in the technology use…”
Line #280 – Based on comments from two reviewers, we changed” the text to: “Based on results published in 33 articles, the systematic review of SFRA literature, previously published in [47], identified evidence of that SFRA can be effective in assessing older adults’ fall risk, both in terms of discriminating groups with varied levels of fall risk from each other and in terms of classifying individuals based on fall risk.” We have also changed “EU project” to “European collaboration project”.
Line #314 –The abbreviations were removed.
Line #324 – Line #332 – we changed to “The participants describe that the patients’ fall risk is assessed in various settings some of which included meetings with out-patients, as well as in-patient enrollment, post-surgery rehabilitation planning etc. They described that they observe and collect information both from the patients and their caregivers/ relatives (Appendix A). They meet a broad range of patients, e.g., patients with injuries caused by falls or other reasons, and patients with a varying level of balance- and walking abilities.”
Line #331 –We believe that the suggestion does not fully express what we wanted to say. The sentence has been rephrased to “Most participants also meet patients with arm injuries inhibiting arm movements”
Line #340 – we changed to ”Some participants were familiar with at least one of the FRA methods used in the SFRA approach selected in sub-study 3.”
Line #351 – we changed to “Sub-study-2 identified three evaluations of two SFRA methods’ abilities to discriminate between groups of patients with different fall risk levels.”
Line #370 – we changed to “The participants’ willingness to dedicate time to SFRA, i.e., to mount and remove sensors, and to review SFRA results, varied (see Figure 7).”
Line #375 –we believe that the second suggestion does not fully express what we wanted to say. Therefore, we kept the sentence “one of them questioned if SFRA should be performed in out-patient care where the scheduled visits are short.”
Line #387 –we changed to anticipated, outcome and value
Line #393 – we changed to “Negative outcomes anticipated included time constraints, cost, waste of resources, as well as insufficient efficacy and reliability”.
DISCUSSION:
Line #409 – Delete in-patient and just say during hospital stay, it is in the beginning of the sentence.
Line #422 – not sure if you need to repeat reference #63 again since you just listed it above.
Line # 425 – Fall risk(s) (may be multiple) and classify them according to the risk(s).
Line #432 – Line #433 – In order to not be repetitive again, consider revising to: collecting data in semi-supervised or supervised environment and only include one or two assessment tasks.
Line #442 – Change “higher” to “larger”.
Line #453 – Change “hinder” to “impact”.
Line #455 – change ‘hinder” to “barrier”.
Line #456 – Line #457 – It was emphasized that SFRA systems need to be easy to use, work smoothly and be reliable to eliminate the stress among staff as well at the patients. Delete sentence in Line #458.
Line #462 – Change the end of the sentence to accept SFRA and its value.
Line #466 – Line #473 – SFRA has been repeated multiple times and can be eliminated easily.
Authors response:
The following changes have been implemented according to the reviewer’s comments:
Line #409 – Delete in-patient and just say during hospital stay, it is in the beginning of the sentence.
Line #422 – not sure if you need to repeat reference #63 again since you just listed it above.
Line #442 – Change “higher” to “larger”.
Line #455 – change ‘hinder” to “barrier”.
The following comments from the reviewer has been addressed as follows:
Line # 425 – we changed to “levels of fall risk and classify older adults according to risk levels”
Line #432 – Line #433 – we changed to “collecting data in semi-supervised or supervised FRA settings only include one or two assessment tasks.”
Line #453 –we believe that the second suggestion does not fully express what we wanted to say. Therefore, we kept the word hinder here.
Line #456 – Line #457 – we believe that the second suggestion does not fully express what we wanted to say. Therefore, we have changed to “It was emphasized that SFRA systems need be easy-to-use, work smoothly and be reliable to prevent stress among the staff. System reliability has also been identified as important for older adults’ adherence to self-assessment of fall risk [44].
Line #462 – Changed to “accept SFRA and its outcomes”
Line #466 – Line #473 - This text has been re-formulated to eliminate repeating the word SFRA.
CONCLUSIONS:
Reviewer’s comments:
Line #477 – Revise to: FRA methods that may be relevant in orthopedic clinics. Line #480 – Delete “and” before reliable.
Authors response:
Suggested change has been implemented.
TABLES:
Reviewer’s comments:
Overall comments – If possible, consider separating the table title (in the top) and the other information such as descriptions or abbreviations at the bottom. Too much information and very difficult to follow. Not sure if some can be eliminated.
Authors response:
A large part of the abbreviations has been eliminated. However, as we are following the journal template, the descriptions and abbreviations are still in the top of the tables.
TABLE #1:
Reviewer’s comments:
Per title, situations to explore are highlighted in Italics, did not see any italicized sections.
Row #3, Column #4 – This research is completed, so all the information should be past tense….Mobility was evaluated, etc.
Row #5, Column #4 – Delete “most participants were patients that have … this is already listed under aim column.
Row #6, Column #2 – Revise to Identification and recommendations for home adjustments, ADL training, ….
Authors response:
The following changes have been implemented according to the reviewer’s comments:
Row #5, Column #4 – Delete “most participants were patients that have … this is already listed under aim column.
The following comments from the reviewer has been addressed as follows:
Row #3, Column #4 – The texts in column 4 describe work methods in a clinic (not completed research). We have therefore changed the sentences that used verbs to instead use nouns.
Row #6, Column #2 – Revised to ” dentification and recommendations for adjustments and municipal rehabilitation.….”
TABLE #2:
Reviewer’s comments:
For titles, the first column can just be titled as Study/year
Under study population, ref. #51, you listed patients (geriatric). A little confusing since you have used “older adults” in the text, how is this population different?
Authors response:
The first suggestion has been implemented. For the second one, we have clarified that the participants were inpatients in a geriatric clinic.
TABLE #3:
Reviewer’s comments:
Only comment here will be the general as above and if you consider the reviewer comments, RN instead of N for nurse.
Authors response:
See response to general comments above, abbreviation for nurse has been removed from table.
FIGURESS:
Reviewer’s comments:
No comments!!
Authors response:
No changes performed.
Reviewer 2
Interesting study that aims to understand the acceptance and applicability of using sensor-based in fall risk assessment.
Because it is a study carried out in several phases (and with results already published in other articles), there is information in this article that cannot be understood without reading the other articles.
It would be advisable to briefly explain what was presented in these articles and what is relevant to understanding this particular article.
For example:
P5 L178-180
“Methods for literature searches, literature screening, data abstraction, analysis of data on SFRA methods and their performance in FRA are described in [47].”
- The sentence does not add information to the study. The analysis of relevant data and their performance in FRA should be mentioned here.
Direct references to other articles without explanation of their relevance should be removed. For example: “Examples of studies on acceptance and experiences of SFRA among independently living older adults are [44] and [45].” - The sentence does not add information to the study.
It is suggested: There are already studies on the acceptance and experiences of SFRA among elderly people with independent living [44, 45], so it is pertinent to carry out this type of study in hospitalized elderly people.
P 10 L280
“The results of the systematic literature review of SFRA were published in [47].” – if there is important information, it should be described in this article. This sentence does not add information to the study.
P13 L353-356
“Despite not being considered as relevant for clinical FRA according to the criteria in sub-study 3, the researchers were interested in whether the clinical staff working at the orthopedic clinic found these FRA methods, i.e., elbow flexion test and 6-Minutes-Walking-Test (6MWT), potentially relevant.”
I could not understand where the reference to these assessment instruments comes from (by reading this article exclusively). It must be explained.
Although the article is based on the analysis of Sensor-Based Fall Risk Assessment, this concept is not explained. There must be a description of this concept and this technology, as well as the necessary material.
Authors response:
We are happy to hear that you find the study to be interesting and thank you for your constructive comment on providing additional information to increase the readability of the article. All your comments have been addressed in the revision, a detailed description on how this has been implemented in the text is given below:
P5 L178-180: Here we have added information on how the literature study was performed: ”The systematic literature review, previously published in [47], was performed by searching in four databases. The studies were selected systematically according to the set egilibility criteria. Data was extracted using a study specific template with defined variables. The aim of the data analysis was to investigate whether there was evidence of SFRA in terms of discriminative capacity and classification performance, and whether previously identified risk factors for study bias could be identified among the included studies.”
The suggestions on how to explain relevance of references have been implemented:
“Examples of studies on acceptance and experiences of SFRA among independently living older adults are [44] and [45].” has been changed to “As, studies on acceptance and experiences of SFRA have been performed among independently living older adults [44, 45], it is pertinent to perform the same type of studies with staff involved in hospital care of older adults.”
P10 (previously L280) has been changed to: “Based on results published in 33 articles, the systematic review of SFRA literature, previously published in [47], identified evidence of that SFRA can be effective in assessing older adults fall risk, both in terms of discriminating groups with varied levels of fall risk from each other and in terms to classify individuals based on fall risk.”
P13 L353-356: Brief explanations of the assessment instruments and references to SFRA-studies which include explicit descriptions of how the instruments have been used have been added.
SFRA has been described on P2 with a reference to a comprehensive review of SFRA-literature: “This has generated an interest in sensor-based FRA (SFRA) which base the assessments on signals from wearable sensors monitoring an individual’s motions while performing specific assessment tasks [32].
Reviewer 2 Report
Interesting study that aims to understand the acceptance and applicability of using sensor-based in fall risk assessment.
Because it is a study carried out in several phases (and with results already published in other articles), there is information in this article that cannot be understood without reading the other articles.
It would be advisable to briefly explain what was presented in these articles and what is relevant to understanding this particular article.
For example:
P5 L178-180
“Methods for literature searches, literature screening, data abstraction, analysis of data on SFRA methods and their performance in FRA are described in [47].” - The sentence does not add information to the study. The analysis of relevant data and their performance in FRA should be mentioned here.
Direct references to other articles without explanation of their relevance should be removed.
For example:
“Examples of studies on acceptance and experiences of SFRA among independently living older adults are [44] and [45].” - The sentence does not add information to the study.
It is suggested:
There are already studies on the acceptance and experiences of SFRA among elderly people with independent living [44, 45], so it is pertinent to carry out this type of study in hospitalized elderly people.
P 10 L280
“The results of the systematic literature review of SFRA were published in [47].” – if there is important information, it should be described in this article. This sentence does not add information to the study.
P13 L353-356
“Despite not being considered as relevant for clinical FRA according to the 353 criteria in sub-study 3, the researchers were interested in whether the clinical staff work- 354 ing at the orthopedic clinic found these FRA methods, i.e., elbow flexion test and 6 - 355 Minutes-Walking-Test (6MWT), potentially relevant.” – I could not understand where the reference to these assessment instruments comes from (by reading this article exclusively). It must be explained.
Although the article is based on the analysis of Sensor-Based Fall Risk Assessment, this concept is not explained. There must be a description of this concept and this technology, as well as the necessary material.
Author Response

(The authors gave the same response as above.)

Round 2
Reviewer 1 Report
Thank you for accepting the comments and applying the suggested changes. I have few very minor corrections as well for your consideration.
Good luck with publication.

Author Response
RESPONSE TO REVIEWERS 
Dear Editor and Reviewers, 
Thank you for your valuable comments in the second revision of the manuscript “Clinical Sensor-Based Fall Risk Assessment at an Orthopedic Clinic: A Case Study of the Staff’s views on Utility and Effectiveness” (revised title).
Your comments have resulted in several improvements. Below follows point-by-point information on how the comments from each reviewer have been addressed in this major revision of the manuscript. In addition, we have checked the English language and modified the layout of Figure 6 to make it more in similar toFigure 7. Changes made within the manuscript are indicated by track changes.  
Please note that the numbering of lines in the edited manuscript differ from the numbering of lines in comments from Reviewer 1. In one case, we have made revisions on lines additional to what the reviewer suggested. To describe those revisions, we have used the number of lines in the edited manuscript (for example: “Line # 471 and Line # 873 in the edited manuscript: changed to (Figure 1)”.
Reviewer 1 
Reviewer’s comment:
- General Comments
Thank you for accepting the comments and applying the suggested changes. I have few very minor corrections as well for your consideration. Good luck with publication.
- Specific Comments
Thank you to the authors for applying the revision comments for improving the manuscript flow. Below I just have few more minor suggestions for your consideration. Good luck and great job.
Line #73 - Should read: focuses on design and prediction of adaptation of new technology; delete technology acceptance!! your readers will know what you mean by the sentence.
Change word” adopt” to “adapt”.
Line #113 – FRA in older adults, delete “of”
Results
It is okay to keep the abbreviations such as PT, OT, MD, RN, etc. after you have spelled them out. I was just questioning the appropriate abbreviation for doctor and nurse. Sorry for my misleading comment… This should apply to the table as well. If not changed, no problem….
Line #262 – Delete “see” in the (See Table 1).
Line #331- #332 – Delete “and daily life movement (gait and activity)” and just say “Gait”. Also, thanks for correcting the “between one and five participants” it is still not correct (one-five)? Are you trying to say “only one out of five participants were familiar with each FRA method used”?
Line #346 – Please correct “Noone”, I think the space was accidently removed: “No one”
Line #467 – Delete “and” before “reliable”.
Authors response:
We thank the reviewer for compliments on the manuscript and the suggestions on minor corrections. Your comments have resulted in further improvements. Below follows point-by-point information on how the comments have been addressed in this minor revision of the manuscript:
Line #73: We have changed to: “focuses on design and prediction of adoption of new technology”.
We believe that “adoption” (the action or fact of choosing to take up, follow, or use something) is appropriate to use here, while adaptation (the process of modifying something) is not. Hence, we have used the word adoption here.
Line #113: We have changed to “FRA in older adults”
Results
We chose to not use the abbreviations in the text nor table.
Line #262: We have changed to (Table 1). In addition, we removed “see” on other references to tables and figures in the text:
Line # 471 and Line # 873 in the edited manuscript: changed to (Figure 1)
Line # 885 in the edited manuscript: changed to (Table 2)
Line # 921 in the edited manuscript: changed to (Figure 2)
Line # 1502 in the edited manuscript: changed to (Figure 5)
Line # 1635 in the edited manuscript: changed to (Figure 6)
Line # 1643 in the edited manuscript: changed to (Figure 7)
Line #331- #332 – Delete “and daily life movement (gait and activity)” and just say “Gait”. Also, thanks for correcting the “between one and five participants” it is still not correct (one-five)? Are you trying to say “only one out of five participants were familiar with each FRA method used”?
We changed to “gait and activities in daily life” (we do not think that this is the same thing as just “gait” here).
The sentence “One-five participants were familiar with each FRA method.” Has been changed to “Activities in daily life was more well-known among the participants than the other FRA methods”
Line #346: We changed to “no one”
Line #467: We delete the “and” before “reliable”.
Reviewer 2 
Reviewer’s comment:
Congratulations on the correction. The requested clarifications were made and the language used improved. It is ready to be published.
Authors response:
We thank the reviewer for compliments on the manuscript.

Reviewer 2 Report
Congratulations on the correction. The requested clarifications were made and the language used improved.
It is ready to be published.
Author Response

(The authors gave the same response as above.)
